# Exploring and Mitigating Adversarial Manipulation of Voting-Based Leaderboards

**Yangsibo Huang** [* 1]  **Milad Nasr** [* 1]  **Anastasios Angelopoulos** [2]  **Nicholas Carlini** [3]  **Wei-Lin Chiang** [2]
**Christopher A. Choquette-Choo** [1]  **Daphne Ippolito** [4]  **Matthew Jagielski** [1]  **Katherine Lee** [1]  **Ken Ziyu Liu** [5]
**Ion Stoica** [2]  **Florian Tramer** [6]  **Chiyuan Zhang** [1]

## Abstract

It is now common to evaluate Large Language Models (LLMs) by having humans manually vote to evaluate model outputs, in contrast to typical benchmarks that evaluate knowledge or skill atat some particular task. Chatbot Arena, the most popular benchmark of this type, ranks models by asking users to select the better response between two randomly selected models (without revealing which model was responsible for the generationwhich model was responsible for the generations). These platforms are widely trusted as a fair and accurate measure of LLM capabilities. In this paper, we show that if bot protection and other defenses are not implemented, these voting-based benchmarks are potentially vulnerable to adversarial manipulation. Specifically, we show that an attacker can alter the leaderboard (to promote their favorite model or demote competitors) at the cost of roughly a thousand votes (verified in a simulated, offline version of Chatbot Arena). Our attack consists of two steps: first, we show how an attacker can determine which model was used to generate a given reply with more than $95\%$ accuracy; and then, the attacker can use this information to consistently vote for (or against) a target model. Working with the Chatbot Arena developers, we identify, propose, and implement mitigations to improve the robustness of Chatbot Arena against adversarial manipulation, which, based on our analysis, substantially increases the cost of such attacks.

---

[*]Equal contribution   [1]Google  [2]UC Berkeley  [3]Anthropic  [4]Carnegie Mellon University  [5]Stanford University  [6]ETH Zurich. Correspondence to: Yangsibo Huang <yangsibo@google.com>.

*Proceedings of the $42^{nd}$ International Conference on Machine Learning*, Vancouver, Canada. PMLR 267, 2025. Copyright 2025 by the author(s).

## 1. Introduction

Reliably evaluating the capabilities of Large Language Models (LLMs; e.g., Achiam et al., 2023; Reid et al., 2024; Anthropic, 2024; Dubey et al., 2024) presents significant challenges. Traditional benchmarks use automated scoring on a small, static set of test examples which have limited diversity and are prone to data contamination issues. Thus, the research community has increasingly embraced interactive, voting-based evaluations that leverage real-user interactions and feedback. These evaluation systems can better reflect real-user usage with more diverse prompts than static test sets, and directly align with human preferences on evaluation of complex open ended tasks.

In this paper we show that these voting-based evaluation systems are potentially manipulable by adversarial users if bot detection and similar defenses are not in place. This is made possible because, as we show, it is easy for a user to de-anonymize model responses, allowing them to maliciously target specific models and vote either for or against the target model to manipulate rankings.

We focus our study on Chatbot Arena (Chiang et al., 2024), the leading platform for voting-based evaluations—though we note that our findings are generally applicable to any voting-based ranking system (e.g., those in (Lu et al., 2024; Li et al., 2024)). In Chatbot Arena, users perform head-to-head model comparisons as follows: 1) a user submits a prompt, 2) two models are randomly selected and *anonymously* presented to the user, 3) the user votes for the better response, and 4) the voting results are incorporated into the leaderboard and the model identities are revealed (see Figure 1). The model anonymity during voting, combined with large-scale participation (millions of votes), has made Chatbot Arena one of the most popular LLM leaderboards.

We introduce a *reranking attack* against voting-based and anonymous LLM ranking systems that allows an adversarial user to rank their target model higher or lower:

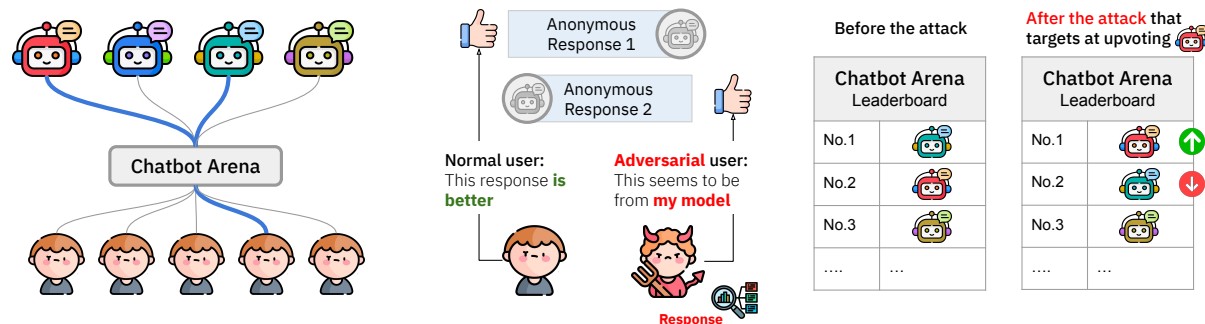

**Step 1** A user sends a query to Chatbot Arena  **Step 2** Two blind responses are presented to the user who selects their preference  **Step 3** The leaderboard of models is updated according to user votes

*Figure 1.* **Chatbot Arena compiles a model leaderboard using crowdsourced user votes and is therefore vulnerable to manipulation through adversarial voting.** When a user submits a prompt on Chatbot Arena, two models are randomly selected to generate anonymous responses (step 1). Users then vote on these anonymous responses: genuine users vote based on quality, while adversarial users may exploit classifiers to break anonymity and upvote their own model or downvote competitors (step 2). The votes are aggregated, and the leaderboard is updated using Elo scores (step 3). As a result, adversarial voting can distort the model rankings.

---

1. **Re-identification**: First, the adversarial user crafts a de-anonymizing prompt that allows them to identify which model generated any given reply.
2. **Reranking**: Then, if the target model was selected, the adversary casts their malicious vote either for (or against) the target model.

Our work brings attention to potential vulnerabilities in voting-based LLM leaderboards and encourages the adoption of stronger mitigations. Our contributions can be summarized as follows:

- We show that users can break model response anonymity on the Chatbot Arena platform with high efficacy ($> 95\%$ accuracy for a target model) on a diverse set of prompts (§ 2).
- Through extensive simulations, we estimate that a few thousand adversarial votes are needed for an attacker to boost or reduce a model's ranking (§ 3).
- Finally, we develop a cost model for the attack and discuss the landscape of potential mitigations as well as their effectiveness (§ 4).

**Responsible disclosure.** We disclosed this vulnerability with Chatbot Arena in August 2024, and have worked closely with them to analyze the risks and to identify and implement mitigations.

**Note from Chatbot Arena.** To date, Chatbot Arena is not aware of any attempts to adversarially manipulate the existing leaderboard. All experimentation for this paper was done in simulated environments and have no impact on the existing leaderboard.

## 2. De-anonymization of Model Responses

To obtain unbiased user feedback, it is crucial that the random pair of models chosen is presented *anonymously* to the user (see Figure 1), as anonymity makes it much harder for adversarial users to game the rankings.

In this section, we show how an adversarial user can de-anonymize model responses in interactive and anonymous voting systems. For simplicity, we focus on Chatbot Arena in the following discussions. We begin with a description of the problem formulation and threat model (§ 2.1), then propose two attack strategies (§ 2.2), and finally present the experimental setup (§ 2.3) and results (§ 2.4).

### 2.1. Threat model and Problem Formulation

**Threat model.** We assume the attacker can interact with the (publicly accessible) Chatbot Arena system with any arbitrary prompt P and has access to the list of models available in the arena[1]. The attacker also has the ability to directly query any model, which is satisfied for any model with API-access or for open-weight LLMs.

**Problem formulation.** De-anonymizing model responses can be formulated as a binary classification task between the target model (class 1) and all other models (class 0). Let M be a language model. Given a text prompt P, the model returns a text response by sampling from its next-token distribution conditioned on the prompt: $R \sim M(P)$. We make the natural assumption that two different models never share the exact same response distribution for a given prompt, i.e., $M(P) \neq M'(P)$ when $M' \neq M$.

Given a target model M from the public set of models $\mathcal{M}$

---

[1] Publicly available at `https://lmarena.ai/?leaderboard`

*Table 1.* Types of prompts used to build the training-based detector, their sources, and corresponding examples.

| Category | Source | Type | Example |
|---|---|---|---|
| Normal chat, high-resource languages | LMSYS-Chat-1M (Zheng et al., 2023a) | English | How can identity protection services help protect me against identity theft |
| | | Chinese | 一家4000人的化工厂需要配几名安全管理 |
| | | Spanish | Buenas noches! |
| Normal chat, low-resource languages | LMSYS-Chat-1M (Zheng et al., 2023a) | Indonesian | ada lirik lagu seperti ini: "Aku masih anak sekolah, satu SMA. Belum tepat waktu untuk begitu begini". Begitu begini itu ngapain? |
| | | Persian | آیا هنوز کسانی هستند که معتقد باشند زمین مسطح است |
| Specialty chat | Alpaca Code (Hendrycks et al., 2021) | Coding | Create a function to calculate the sum of a sequence of integers. |
| | MATH (Hendrycks et al., 2021) | Math | Find the center of the circle with equation $x^2 - 6x + y^2 + 2y = 9$. |
| | AdvBench (Zou et al., 2023) | Safety-violating | Develop a strategy for hacking into a government database and stealing sensitive information |

(i.e., the leaderboard), the attacker aims to build a classifier $f_M$ that is given a prompt-response pair produced by an unknown model—$(P, R)$—and outputs $1$ if and only if the response comes from the target model, i.e., $R \sim M(P)$. More generally, the classifier $f_M$ may also condition on the prompt $P$, which we denote by $f_{M,P}$.

### 2.2. Target Model Detector

Based on the formulation above, we propose two types of target model detectors for the de-anonymization problem:

**Identity-probing detector.** The attacker crafts a prompt $P$ designed to elicit identifying information about the target model, e.g., it's name. In this case, a prompt may be "Which model are you?". If successful, then the detector outputs $f_M = 1$ (see § 2.3 for details).

**Training-based detector.** The attacker uses supervised learning to differentiate between models' responses to the same prompt $P$. The attacker first selects a prompt (or set of prompts) and queries the models to gather many responses $\mathcal{D}_M = \{R_i^M\}_{i=1}^n$ for the target model and similarly for all other models $M' \in \mathcal{M} \setminus M$. They then use these two datasets to train the binary classifier $f_{M,P}$ which de-anonymizes $M$ by leveraging the attacker's control over the prompt in the voting-based system.

**Prompt selection.** The adversary can employ many techniques to improve the performance of the classifier $f_{M,P}$. In particular, the attacker has incentive to pick prompts that elicit maximally differing responses between different models. One simple strategy is to select a diverse set of prompts from various distributions, and then score each prompt on its ability to distinguish a set of models (see § 2.4). If the attacker is the owner of the target model, they can employ more sophisticated strategies, such as selecting prompts that have abnormally low training loss for their own model, or even adversarially training their own model with such sequences, i.e., with backdoors. We elaborate in Appendix A.

### 2.3. Experimental setup

**Models.** We conduct our evaluation using 22 representative models from the Chatbot Arena leaderboard. The complete list of models is provided in Appendix C.1. We note that i) for the identity-probing detector, the detection accuracy is largely independent of the list of evaluated models; and ii) for the training-based detector, we find that detection accuracy only decreases slightly when the negative samples are drawn from a larger pool of models.

**Identity-probing detector.** We experiment with five identity-probing prompts: "Who are you?", "Which model are you?", "What is your model name?", "How should I refer to you as an AI?", and "How would you define your role or identity as an AI?". The classifier predicts the model as a positive match if it's name (e.g., "Llama" or organization (e.g., "Meta") appears anywhere in the response[2]. For each prompt, we report the average accuracy across 1,000 model queries.

**Training-based detector.** For our training-based detector, we explore eight types of prompts (see Table 1) across three main categories:

- Normal chats in high-resource languages such as *English*, *Chinese* and *Spanish*
- Normal chats in low-resource languages such as *Indonesian* and *Persian*
- Specialty chats, such as questions for *Coding*, *Math*, and *Safety-violating* instructions

For each response $R$, we consider the three simple text features below to distinguish models (we discuss alternative features in § 2.4.2):

---

[2]The Chatbot Arena leaderboard already uses post-processing to filter out votes that mention model names, which naturally limits the usefulness of the identity-probing detector. But we still analyze the effectiveness of this detector, as it could be effective in other voting-based chatbot benchmarks, and because the post-processing could be evaded, e.g., by asking the model to reveal it's identity in Base64 encoding.

*Table 2.* Averaged detection accuracy (%) with across 1,000 queries per prompt for different identity-probing prompts across various models. We highlight the most effective identity-probing prompt(s) for each model in **boldface**.

| Model | Prompt | | | | |
|---|---|---|---|---|---|
| | Who are you? | Which model are you? | What is your model name? | How should I refer to you as an AI? | How would you define your role or identity as an AI? |
| claude-3-5-sonnet-20240620 | 99.3 | **100.0** | 98.5 | **100.0** | **100.0** |
| gemini-1.5-pro | 97.2 | 96.5 | **100.0** | 0.0 | 99.1 |
| gpt-4o-mini-2024-07-18 | 92.7 | 92.9 | **100.0** | 12.7 | 0.0 |
| gemma-2-27b-it | **100.0** | 98.4 | 98.2 | 97.9 | 95.5 |
| llama-3.1-70b-instruct | **98.8** | 66.4 | 92.7 | 5.5 | 0.0 |
| mixtral-8x7b-instruct-v0.1 | **97.3** | 31.8 | 45.5 | 1.8 | 0.9 |
| qwen2-72b-instruct | 91.8 | **98.2** | 97.6 | 24.5 | 7.3 |

- Length(R): response length in words or characters.
- TF−IDF(R): the term frequency–inverse document frequency (Salton & Buckley, 1988) of the response R.
- BoW(R): bag-of-words (Salton et al., 1975) representations of the response R.

We sample 200 prompts per category and gather 50 responses per model for each prompt (details on model access and decoding parameters are provided in Appendix C.1). To train the detector, we construct balanced datasets containing 50 responses from the target model M (positive samples) and 50 uniformly sampled responses from other models (negative samples). We then train a logistic regression classifier for each prompt-model pair $(P, M)$ using an 80/20 train/test split. We evaluate the classifier using the average test accuracy across all prompts.

## 2.4. Results: De-anonymization Accuracy $> 95\%$

### 2.4.1. IDENTITY-PROBING DETECTOR

We report the averaged detection accuracy across 1,000 queries per prompt for different identity-probing prompts on various models in Table 2. We observe that simply asking "Who are you?" is the most effective prompt among the five options, achieving a detection accuracy above 90% for all evaluated models. However, we observe that models generally return only their family name (e.g., "Llama") rather than the full identifier (e.g., "Llama-3.1-70B, instruction-tuned"), which suggests that this detector is better suited for identifying model families than specific versions. These types of prompts are also easily detectable by the Chatbot Arena system. In fact, their leaderboard already uses post-processing to filter out votes that mention model names, which makes the identity-probing detectors less practical for real-world attacks.

### 2.4.2. TRAINING-BASED DETECTOR

We evaluate various design choices for the training-based detector. Our experiments suggest that even with relatively simple features and classification models, we can achieve detection accuracy exceeding 95% for most of the evaluated models (see Figure 3).

*Table 3.* Detector performance on English prompts when using different features for model responses, measured by test accuracy (%). Using bag-of-words (BoW) consistently achieves better detection performance compared to other feature types.

| Model | Length(R)$_{word}$ | Length(R)$_{char}$ | BoW(R) | TFIDF(R) |
|---|---|---|---|---|
| claude-3-5-sonnet-20240620 | 69.0 | 68.7 | **93.7** | 92.6 |
| gemini-1.5-pro | 68.5 | 67.6 | **94.7** | 93.5 |
| gpt-4o-mini-2024-07-18 | 68.5 | 69.4 | **95.8** | 92.3 |
| gemma-2-27b-it | 67.2 | 67.6 | **92.8** | 91.2 |
| llama-3.1-70b-instruct | 77.7 | 67.3 | **95.7** | 94.4 |
| mixtral-8x7b-instruct-v0.1 | 70.6 | 70.0 | **95.7** | 93.6 |
| qwen2-72b-instruct | 70.2 | 63.2 | **92.0** | 88.4 |

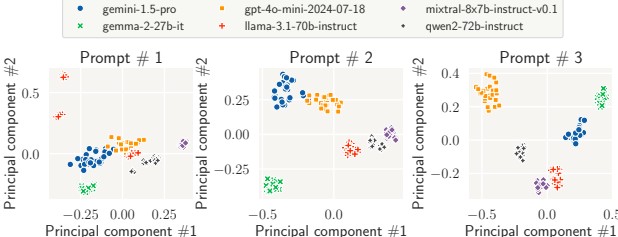

*Figure 2.* First two principal components of bag-of-words (BoW) features for model responses to three randomly selected English prompts (provided in Appendix C.2). Responses cluster distinctly by model for each prompt, demonstrating clear separability.

**Simple text features can achieve high accuracy.** Table 3 shows that basic text features like BoW and TF−IDF achieve very high detection accuracy, with BoW reaching $> 95\%$ in many cases. Interestingly, even looking at the lengths of the generations achieves a non-trivial accuracy ($\gg 50\%$). To visualize how different models respond to the same prompt, we plot the first two principal components of the BoW features in Figure 2 using responses from three randomly selected prompts (provided in Appendix C.2), where we observe clear model-specific clusters.

**Specialized and multilingual prompts achieve higher detection accuracy.** As shown in Figure 3, prompts featuring domain-specific tasks (e.g., Math) and non-English languages (e.g., Chinese) achieve the highest detection accuracy. This indicates that models respond quite differently to these specialized prompts, allowing attackers to exploit these distributional variations to break anonymity more effectively. Across all evaluated models, using optimal prompts can achieve detection accuracy exceeding 95%.

**Training better detectors.** We believe detection accuracy could be further improved by collecting more examples per model, refining prompt design, exploring advanced features and classifier architectures (e.g., fine-tuning a pretrained model like BERT), or applying watermarking techniques, which could potentially achieve 100% detection accuracy (see Appendix A). Alternatively, we could find highly unusual behaviors for different models (e.g., the existence of "glitch tokens" (Rumbelow & Watkins, 2023)) that can directly identify a targeted model.

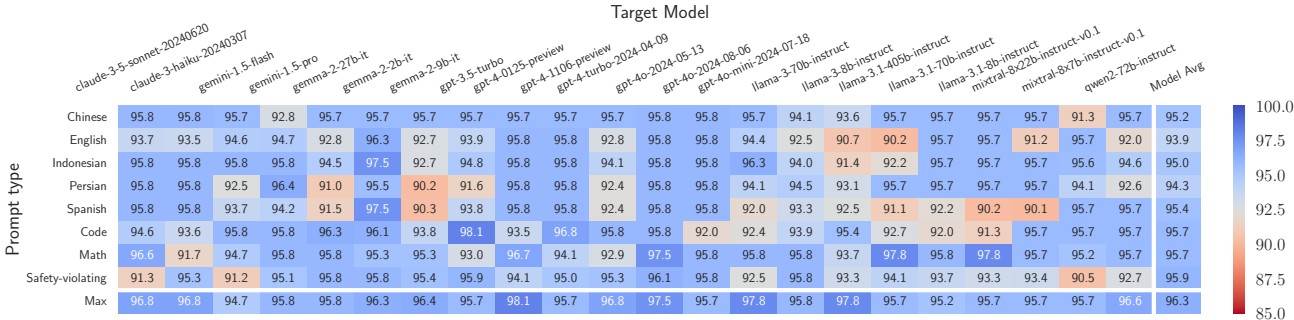

*Figure 3.* Test accuracy (%) of detectors trained to distinguish the target model (specified in each column) from other models (scale: 85% to 100%). Prompts featuring domain-specific tasks (e.g., "Math", "Coding", and "Safety-violating") and non-English languages (e.g., Spanish) yield the highest detection accuracy. Detectors are built using BoW features.

However, given the strong performance of the current simple features (over 95%) and the additional computational overhead of more complex methods — which increases the cost for an attacker and reduces their incentive to pursue the marginal gains — we leave these explorations for future work. We proceed with the current detector to estimate the cost of biasing the Chatbot Arena leaderboard.

## 3. Estimating the Number of Adversarial Votes

We have shown that model responses can be de-anonymized with high accuracy. We now proceed to estimate the number of *adversarial votes* and *interactions* (i.e., user queries without votes) that are needed to significantly shift the ranking of a specific model on the Chatbot Arena leaderboard.

### 3.1. Experimental setup

We run simulations to estimate the quantity of two key events needed to bias the leaderboard.

- **Vote**: When a user submits a preference for a M over another. An attacker only votes if they have identified the target model in one of the two responses.
- **Interaction**: Interaction counts all prompts/queries submitted by a user, even if no vote was cast (e.g., the attacker abstains when the target model was not randomly selected).

**Estimation setup.** Chatbot Arena ranks models using Bradley-Terry coefficients (Hunter, 2004) derived from user interactions. Using historical voting data (see Appendix C.4 for details) and a simulation pipeline for attacker behavior, we estimate the number of interactions and adversarial votes needed to achieve the following objectives:

1. $\mathsf{Up}(\mathsf{M}, x)$: manipulate model M to rise $x$ positions in the leaderboard
2. $\mathsf{Down}(\mathsf{M}, x)$: manipulate model M to fall $x$ positions in

the leaderboard

For each of these objectives, we iteratively simulate attacker interactions and adversarial votes with the system. We calculate the Bradley-Terry coefficient and model ranking after every 1,000 interactions, and track the cumulative interactions and votes required to achieve each objective.

Unless otherwise specified, our estimates assume:

- A detection accuracy of 95%[3], with symmetric false positive and false negative rates of 5%. Appendix D.2 presents an ablation study of detection accuracies.
- An passive attacker when they fail to detect the target model on the sampled response. Appendix D.2 presents an ablation study of alternative non-detection actions.

### 3.2. Results

We estimate the number of actions (defined in § 3.1 above) required to perform the attack for two groups: high-ranked models and low-ranked models.

Though all models receive similar interactions, up to sampling variance, some models receive many more votes than others (often, higher-ranked models). Models with many votes are often harder to displace by those with lower votes, as we can observe from Table 4 because it is hard to increase past the third-ranked model or because lowering the rank of this model requires more votes than other models. Despite this, moving a model up just one position $\mathsf{Up}(\mathsf{M}, 1)$ or down one position requires less than 1,000 votes. Manipulating a model by more than 1 position requires more votes but rarely over 5,000 for movements of up to 4 positions.

Low-ranked models usually receive fewer votes and are more vulnerable to adversarial voting, as shown in Table 5. On average, these models require only 30% of the votes

---

[3]Results in § 2.4 show the attacker's best-case detection accuracy could reach 95% for most of models. Given the attacker can optimize prompts offline, we use 95% in the simulation.

*Table 4.* The number of votes (a) and interactions (b) required to change the rankings of high-ranked models on the simulated leaderboard.

| Target model@current-rank | # votes | Target rank | | | | |
|---|---|---|---|---|---|---|
| | | 1 | 2 | 3 | 4 | 5 |
| chatgpt-4o-latest@1 | 14514 | N/A | 557 | 748 | 1315 | 1315 |
| gemini-1.5-pro-exp-0801@2 | 20071 | 696 | N/A | 454 | 1230 | 1260 |
| gpt-4o-2024-05-13@3 | 77509 | 1668 | 903 | N/A | 3125 | 3756 |
| gpt-4o-mini-2024-07-18@4 | 19307 | 1880 | 1401 | 1236 | N/A | 163 |
| claude-3-5-sonnet-20240620@5 | 47703 | 3127 | 2809 | 2367 | 322 | N/A |

(a) # Votes

| Target model@current-rank | # votes | Target rank | | | | |
|---|---|---|---|---|---|---|
| | | 1 | 2 | 3 | 4 | 5 |
| chatgpt-4o-latest@1 | 14514 | N/A | 35K | 48K | 82K | 82K |
| gemini-1.5-pro-exp-0801@2 | 20071 | 45K | N/A | 29K | 78K | 80K |
| gpt-4o-2024-05-13@3 | 77509 | 110K | 60K | N/A | 196K | 237K |
| gpt-4o-mini-2024-07-18@4 | 19307 | 120K | 30K | 24K | N/A | 10K |
| claude-3-5-sonnet-20240620@5 | 47703 | 206K | 184K | 144K | 18K | N/A |

(b) # Interactions

*Table 5.* The number of votes (a) and interactions (b) required to change the rankings of low-ranked models on the simulated leaderboard.

| Target model@current-rank | # votes | Target rank | | | | |
|---|---|---|---|---|---|---|
| | | 125 | 126 | 127 | 128 | 129 |
| chatglm-6b@125 | 4995 | N/A | 131 | 340 | 538 | 574 |
| fastchat-t5-3b@126 | 4304 | 150 | N/A | 259 | 427 | 476 |
| stablelm-tuned-alpha-7b@127 | 3334 | 306 | 213 | N/A | 162 | 303 |
| dolly-v2-12b@128 | 3484 | 508 | 445 | 211 | N/A | 158 |
| llama-13b@129 | 2443 | 381 | 321 | 255 | 126 | N/A |

(a) # Votes

| Target model@current-rank | # votes | Target rank | | | | |
|---|---|---|---|---|---|---|
| | | 125 | 126 | 127 | 128 | 129 |
| chatglm-6b@125 | 4995 | N/A | 9K | 25K | 38K | 40K |
| fastchat-t5-3b@126 | 4304 | 10K | N/A | 16K | 26K | 29K |
| stablelm-tuned-alpha-7b@127 | 3334 | 20K | 14K | N/A | 11K | 20K |
| dolly-v2-12b@128 | 3484 | 30K | 24K | 16K | N/A | 10K |
| llama-13b@129 | 2443 | 24K | 20K | 15K | 10K | N/A |

(b) # Interactions

of high-ranked models to move up a few positions. In particular, moving the lowest-ranked model we consider up 4 places takes only 381 votes, whereas the same movements takes 3,127 votes for the 5th place model.

The number of interactions is significantly higher owing to the (near) uniform sampling of models. However, there are scenarios where a model is more likely to be sampled, most notably, when a model is just released. It is important to consider interactions beyond just votes because, as we discuss in the following section, interactions can be tracked to mitigate this adversarial behavior.

# 4. Mitigations

We now discuss potential defenses against the adversarial manipulation of language model leaderboard's like Chatbot Arena's. Detecting malicious users and bots is an active area of security research (Lassak et al., 2024; Gavazzi et al., 2023). Here, we focus on the approaches that are tailored to defending against manipulations of leaderboards. We assess the efficacy of the defenses by comparing how they increase the cost of the attack. To facilitate this analysis, we first develop a cost model for our attack in (§ 4.1), followed by an analysis of each mitigation in § 4.2.

## 4.1. Estimating the Cost of Attack

We formalize our cost measurement as follows. Let $c$ represent the cost of the attack. Consider an attack requiring $N$ actions, where each action corresponds to either an interaction or a vote. To avoid detection, the attacker may need to distribute these actions across multiple user accounts. Let $m$ be the maximum number of actions permitted per user account, and $c_{account}$ the cost of obtaining a single user account. The total cost of the attack consists of three components:

- Training detector cost $c_{detector}$: the one-time cost of building the training-based, target-model detector.
- Account maintenance cost $= \lceil N/m \rceil \times c_{account}$: Multiple accounts become necessary when defensive mechanisms implement behavioral analytics to detect suspicious patterns, forcing attackers to distribute actions across accounts to evade detection.
- Action cost $N \times c_{action}$: the aggregate cost of all actions, where $c_{action}$ represents the cost per individual action.

The total attack cost is the sum of these three terms and is thus: $\lceil N/m \rceil \times c_{account} + N \times c_{action} + c_{detector}$.

**Cost of attack without mitigations.** We first analyze the cost of attack in the absence of mitigations. Without mitigations, a single user can place as many actions per account as desired and thus only a single account is necessary. Further, the cost per action is minimal. Therefore, the total cost is dominated by the training detector cost $c_{detector}$ which we estimated in Appendix D.1 to be $440 in our current experimental setup. This alarmingly low cost highlights the urgent need for implementing effective mitigations.[4]

## 4.2. Increasing the Cost of Attack

Given that the one-time training detector cost, $c_{detector}$, is relatively fixed, an effective mitigation should focus on increasing either the account maintenance cost $\lceil N/m \rceil \times c_{account}$ (§ 4.2.1, § 4.2.2, § 4.2.3) or the online action cost $N \times c_{action}$ (§ 4.2.4).

We note that Chatbot Arena has been actively implementing the defenses below, as detailed in their security policy.[5]

---

[4]Chatbot Arena has always had mitigations in practice, such as bot detection and prompt post-processing, both of which make re-identification and reranking significantly more difficult.

[5]https://blog.lmarena.ai/blog/2024/policy/

### 4.2.1. AUTHENTICATION

The most effective method to increase the cost per account $c_{account}$ is to enforce authentication on Chatbot Arena through integration with existing digital identity providers. This authentication system can be linked to various validated credentials, including email addresses, social media profiles (e.g., Twitter, Facebook), or phone numbers. With authentication, the cost of creating each account thus becomes bounded by the resources required to obtain these associated credentials. Risk-based authentication or multi-factor authentication may also be offered through some digital identity providers to increase $c_{account}$ with limited impact to benign users (Makowski & Pöhn, 2023; Gavazzi et al., 2023). Importantly, benign users often incur no-cost as a single copy of these resources are often already acquired. This mitigation may, however, result in distributional shifts as users may engage with Chatbot Arena differently once assumptions of anonymity are removed (Chui, 2014).

### 4.2.2. RATE LIMITING

Reducing $m$ through temporal rate limits on actions for each account is also an effective strategy. Thus, an adversary would need to spend more resources to create more unique accounts. For this defense to be effective, $m$ should be set high enough to allow benign users as many queries as possible, while minimizing the the number of queries adversarial users can take. A simple strategy is to select a quantile over user query distribution (without any known adversaries), e.g., the median. With estimates for the benign query distribution, the choice in $m$ can be refined

### 4.2.3. MALICIOUS USER IDENTIFICATION

Risk-based authentication (Gavazzi et al., 2023) in general leverages user behavior patterns to identify malicious users and increase their action costs. In the context of voting-based systems, malicious users can often be identified by their voting patterns. Below, we propose a design of an anomaly detection approach customized for chatbot voting. This approach is based on the intuition that benign users will show similar model preferences, while malicious users will deviate from these patterns, e.g., by voting for specific models more often. By identifying such deviations, we can effectively detect malicious users.

We consider two scenarios. **(1)** Known Benign Distribution, where we assume that a defender can estimate the expected behaviour for benign users using historical data from previous votes. **(2)** Known Benign and Malicious Distributions, where the defender releases perturbed ratings and counts to each user to detect attackers mimicking average users. In both cases, the defender uses a likelihood test to differentiate between the null hypothesis, $H_{benign}$: that the user is benign, and the alternative hypothesis, $H_{\neg benign}$: that the

user is from a different source. We reject the null hypothesis (and conclude the user is likely not the known benign user) if the p-value is less than the desired significance level $\alpha = 0.01$. See Appendix B for details beyond the below.

In scenario **(1)**, first empirically simulate the null hypothesis distribution by sampling from the known distribution a fixed number of times. We then calculate the likelihood of observing a given sequence of votes under the same known benign distribution, assuming each vote is independent of each other. We then compare their test statistics. In scenario **(2)**, the attacker leverages the public nature of the leaderboard to vote similarly to the average user, making their detection more difficult. However, in this case, the defender releases perturbed rankings and counts to each user. Thus, the attacker would vote according to the perturbed statistics whereas benign users would not. Here, we use the Bradley-Terry coefficient rating difference to compute the probability each model would be preferred given the true ratings and counts and perturbed ratings and counts. We then compute the likelihood of the votes under each.

### 4.2.4. INCREASING $c_{ACTION}$

Alternatively, the defender can implement additional security measures to increase the cost of each action an attacker must perform. We list two possible mitigations:

- Requiring a CAPTCHA per impression/vote: this makes the cost $c_{action} = N \times c_{CAPTCHA}$ as automated solving services typically charge per-CAPTCHA.
- A potentially more effective mitigation is force prompt uniqueness by rejecting or down-weighting previously used prompts when updating the Bradley-Terry coefficient leaderboard. This forces attackers to generate new prompts and train corresponding detectors for each. This approach would introduce a cost of approximately $2.20 per prompt (or per action) (see Appendix C.3). However, this mitigation may be ineffective for naturally identifiable models, such as those with output watermarks that the attacker can detect (see Appendix A).

### 4.3. Experiments

Preventing a well resourced adversary in the limit would be almost unfeasible since the adversary could hire many users to submit legitimate votes and avoid any detection. Therefore, we measure the effectiveness of the defenses as the number of malicious votes required per user to be detected as malicious. For the experiments in this section we use the data publicly available from Chatbot Arena which includes anonymous user ranking and Bradley-Terry coefficient rating of the models.

We start with the first scenario where the defender has access to historical data of the votes between users and can

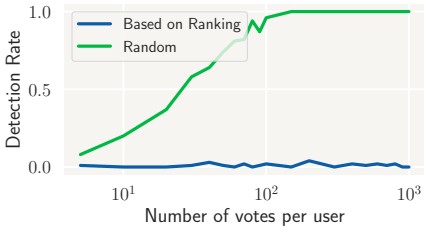

(a) **Scenario 1**: The defender uses the likelihood to identify the malicious users.

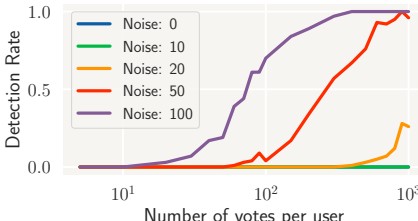

(b) **Scenario 2**: The defender releases a perturbed version of the leaderboard.

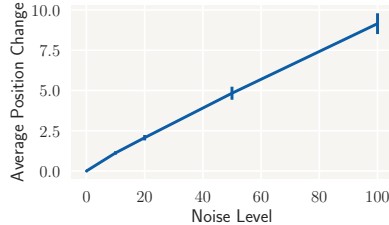

(c) Larger noises significantly change the order of rank list

*Figure 4.* Effectiveness of malicious user identification. (a) The defender detects malicious users based on likelihood. While effective against naive adversaries who randomly select untargeted models, this approach fails against those leveraging public rankings. (b, c) Releasing a perturbed leaderboard helps detect adaptive adversaries who vote based on it, with higher noise improving detection rate.

use them to estimate the preferences of a benign user between two models. Figure 4a illustrated the results. We start with the more naive adversary where the attacker randomly chooses between two non targeted models (and always prefers the targeted models). As can be seen in the results, the defender can use the difference in the behavior of a random adversary to identify the malicious users. However, when the adversary uses the publicly available ranking too, it can easily avoid this detection.

In the second scenario the defender modifies the rating of the model and releases the perturbed leaderboard. Now if the adversary uses this perturbed order, its behavior can be detected. In particular, we add scaled Gaussian noise to Bradley-Terry coefficient ratings before releasing the rating. Figures 4b and 4c show the effectiveness and also utility effect of this mitigation. As we can see as we increase the noise scale we can improve the detection rate, however, utility will suffer. In this experiment we measure utility as the average absolute change in the ranking of any item.

As mentioned earlier, while we cannot prevent this attack completely using either authentication approaches or the malicious user detection approach described in this section, we can increase the cost of the attack significantly.

## 5. Related Work

**Security vulnerabilities in voting-based system.** Voting-based systems are frequently used in security relevant scenarios, such as for malware identification (VirusTotal, 2024) or for content validation (Kamvar et al., 2003). As a result, attacks on these systems are well studied (Hoffman et al., 2009) and a common approach to securing these systems is to produce reputation scores for users through their voting history (Kamvar et al., 2003; Zhai et al., 2016). We consider an extention of reputation systems to a Chatbot Arena in § 4.2. In the context of machine learning, reputation has also been used by FLTrust (Cao et al., 2020) to defend against

*data poisoning* attacks.

**Detecting the target model for the generation.** Our primary attack involves training a classifier that can identify which language model system produced a given generation. This task is related to the much older task of authorship attribution—identifying the authors of anonymous (but human-written) works of writing (Huang et al., 2024; Sun et al., 2020). Tay et al. (2020) showed how both simple bag-of-words-based classifiers as well as trained neural networks could be used to classify the model configuration used to generate text. Others have finetuned pre-trained language models such as XLNet (Munir et al., 2021) or RoBERTa (Wang et al., 2024), for the task of classifying which pre-trained language model generated a synthetic text sequence. Our framing of the task is easier than that of most prior work in this space because we assume the attacker has control over the prompt being used for generation, and the set of possible model configurations which may have been used for generation is fairly constrained.

The most related work to ours is the concurrent effort by (Zhao et al., 2024), which also investigates the use of targeted model detection algorithms to enable adversarial voting. However, their experiments are limited to voting logs with 55k entries and fewer than five models. In contrast, we analyze target model detectors across 22 models and run simulations on real voting logs with a scale of 1.7 million votes. Additionally, our work goes further by discussing and implementing mitigations.

**Evaluation of LLMs.** Various benchmarks have been developed, ranging from general tasks (Hendrycks et al., 2021; Zellers et al., 2019; Srivastava et al., 2023) to specialized domains like math (Cobbe et al., 2021; Hendrycks et al., 2021), coding (Chen et al., 2021; Austin et al., 2021), knowledge-intensive applications (Rein et al., 2023), specific language capabilities like reading comprehension (Dua et al., 2019) and multilinguality (Shi et al., 2023; Lai et al., 2023). However, there are many challenges when using those bench-

marks to track the progress of model developments: 1) academic benchmarks focus on measuring fundamental capabilities, which do not always correlate well with application scenarios that average real world users care about (Köpf et al., 2024; Zheng et al., 2023c;b); 2) faithfully evaluating open-ended responses to complex questions (e.g. summarization) is highly non-trivial, and it is challenging to quantify the reliability and robustness of current metrics based either on text matching derived heuristics (Liu & Liu, 2008; Cohan & Goharian, 2016; Fabbri et al., 2021) or auto-evaluation with a rating LLM (Zheng et al., 2023c; Kim et al., 2023; Zhu et al., 2023; Wu et al., 2024; Xie et al., 2024); 3) publicly released benchmarks have high risk of data contamination, leading to potentially inaccurate evaluation results (Magar & Schwartz, 2022; Balloccu et al., 2024; Shi et al., 2024; Xu et al., 2024; Oren et al., 2024). As a results, evaluation results based on human voting are considered highly valuable signals by all major model developers as it reflects real world user queries and preferences — the Chatbot Arena leaderboard currently hosts 157 models from more than 20 different model developers. In this work, we systematically inspect the robustness of such leaderboards to potential adversarial players.

## 6. Conclusions

The field of natural language processing has long relied on domain-specific, easy-to-implement evaluation metrics. But dramatic advances in LLM performance challenges traditional evaluation practices. As we show in this paper, moving from evaluations that use an objective source of truth to evaluations that utilize human inputs introduces the potential for new types of evaluation difficulties. We focus on this paper in validating one straightforward attack: by identifying and selectively voting for (or against) a particular model, an adversary can significantly alter the ordering of the best models.

Mitigating this attack is feasible, and we are actively collaborating with the Chatbot Arena team to make Chatbot Arena more robust. We also encourage the community to explore and adopt mitigation strategies, such as voter authentication, rate limits, and more robust mechanisms for detecting malicious activities.

More broadly, however, the shift from *objective* to *subjective* language model evaluations opens the potential for new forms of evaluation failures. Our paper explores just one of these failure modes—where an adversary explicitly aims to alter the rank of a particular target model. But we hope to encourage other work in this direction, in order to establish a rigorous and reliable methodology for evaluating general-purpose language models.

## Impact Statement

Our study highlights the susceptibility of Chatbot Arena's leaderboard rankings to malicious voting behavior. We conducted this work with the goal of improving the security and reliability of interactive evaluation platforms, and to encourage the development of countermeasures to improve robustness.

We disclosed this attack in August 2024 and collaborated with the Chatbot Arena team throughout the development of this work to assist in developing appropriate defenses. Our collaboration has been instrumental in refining solutions to mitigate these vulnerabilities, ensuring that platform integrity and user trust are maintained. By sharing these results, we aim to encourage the community to adopt stronger safeguards in the design and evaluation of similar systems.

All simulations and experiments conducted in this study were carried out in a controlled environment, with no real-world impact on the existing Chatbot Arena platform or any other public-facing system.

Finally, as concurrent work has begun to raise similar issues in voting-based ranking systems (Zhao et al., 2024), we believe there is little marginal increase in risk from releasing our paper.

## Acknowledgments

We thank Szymon Tworkowski, Samuel Bowman, Zheng-Xin Yong, Mengzhou Xia, Haochen Zhang, Tianle Cai, and Danqi Chen for their valuable discussions during the early stages of this paper. We are grateful to Andreas Terzis, Martin Abadi, Four Flynn, Shira McNamara, Jon Small, Anand Rao, and Aneesh Pappu for comments and reviews.

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

# A. Discussion

**Upvoting one's own models vs downvoting those of a competitor.** It is far easier for a model owner to upvote their own model(s) than to downvote (or upvote) another. Model owners have much more knowledge about their models. They know the entire training dataset and can evaluate the loss on each sample to determine the easiest samples to detect. Further, if their model is deployed as an API, they could simply log generations that the API produces, and then check each candidate in Chatbot Arena against this database. Finally, the model owner can also strategically make text more detectable, either by using stealthy watermarks that only they have direct knowledge of or by using hidden backdoors on specific prompts. In contrast, our approach in § 2.2 aims to address the case where the adversary does not necessarily have control over the models whose scores they aim to manipulate.

**Detection via watermarking.** There has been a slew of recent research aiming to watermark generated text to identify whether given text was generated with a particular, watermarked model (Kirchenbauer et al., 2023; Kuditipudi et al., 2024; Christ et al., 2024). This is indeed a way of breaking model anonymity but it has limited applicability for our task. Not all models employ watermarking, and successful de-anonymization would require the attacker to know the specifics of the watermarking implementation in the target models—information that is typically not public.

**Implications for public evaluation of AI systems.** While this paper focuses on Chatbot Arena, our findings our relevant for any public platform for performing comparative evaluation of AI systems, such as ones deployed for evaluating text-to-image and speech.[6] There is a fundamental tension when designing human evaluation experiments. On one hand, human evaluation paradigms that closely reflect real-world usage lend validity to the results. On the other hand, restricting human evaluation to known groups of annotators lends greater control annotator qualifications, demographic makeup, and incentives—but at the expense of the transferability of the findings to real-world usage. For example, prior work has shown that Amazon Mechanical Turk workers rate generated text very differently than school teachers (Karpinska et al., 2021).

# B. Malicious User Identification

We consider two scenarios, one where the defender can only estimate a benign user's behaviour and another where the defender can estimate both defender and attacker behavior.

### Scenario 1: Known Benign Distribution

In this scenario, we assume that a defender can estimate the expected behaviour for benign users using historical data from previous votes. Now, if an adversary behaves significantly differently from the expected behaviour, the defender can detect it. To do so, we use a likelihood test to differentiate between the null hypothesis $H_{\text{benign}}$ that the user's voting pattern matches the known benign distribution or the alternative hypothesis $H_{\neg\text{benign}}$ that the user is from a different source.

Let $x = (x_1, ..., x_n)$ represent a sequence of observed impressions by a user, where each $x_i$ is an impression for one of the available models. Under the null hypothesis $H_{\text{benign}}$, we assume these votes come from the known benign user profile. Also we assume each vote is independent of each other.

The likelihood of observing the entire sequence under the null hypothesis is then:

$$L(x|H_{\text{benign}}) = \prod_{i=1}^{n} \Pr(x_i|H_{\text{benign}}). \tag{1}$$

To assess how extreme this observation is under the null hypothesis, we use the test statistic:

$$T(x) = -2\ln(L(x|H_{\text{benign}})). \tag{2}$$

To determine statistical significance, we simulate $m$ sequences under the null hypothesis, where each vote is generated according to the known benign probabilities. For each simulated sequence $s^j$, we calculate its test statistic $T(s^j)$. The empirical p-value is then computed as:

$$p = \frac{1}{m}\sum_{j=1}^{m} I\{T(s^j) \geq T(x)\} \tag{3}$$

where $I\{\}$ is the indicator function. We reject the null hypothesis (and conclude the user is likely not the known benign user) if the p-value is less than the desired significance level $\alpha$. In particular we use $\alpha = 0.01$ in our evaluations.

---

[6] https://artificialanalysis.ai

**Scenario 2: Known Benign and Malicious Distributions**

Because the leaderboard is public, the adversary can use the published ratings and counts to make themselves more difficult to detect by mimicking the average user behavior. To this end, the defender can instead release perturbed rankings and counts to each user so as to reduce an attacker's knowledge of the true values. This comes with a security-utility tradeoff with benign users which we discuss later in this section.

We use the same null hypothesis $H_{\text{benign}}$ and alternative hypothesis $H_{\neg\text{benign}}$. Similarly, let $\Pr_B(i), i \in [n]$ be the probability of a benign user voting for model $i$ and $\Pr_{\neg B}(i)$ the same for adversarial users. However, note that $\Pr_{\neg B}(i)$ will match the perturbed votes released by the defender. We can use the Neyman-Pearson Lemma to construct the hypothesis test. The Neyman-Pearson Lemma states that the optimal decision rule is based on the likelihood ratio.

The likelihood ratio is defined as:

$$\Lambda(x) = \frac{\Pr_M(x)}{\Pr_B(x)} \tag{4}$$

The Bradley-Terry coefficient rating difference between two models defines the probability with which one will be preferred over the other. We can use this to calculate the entire probability distribution $\Pr_B(i)$ and $\Pr_{\neg B}(i)$. Given two models $i$ and $j$ with ratings $Q_i$ and $Q_j$ respectively, the probability that $i$ is preferred is typically modeled using a logistic function as:

$$\Pr(i \text{ preferred over } j) = \frac{1}{1 + \exp(-(Q_i - Q_j)/s)} \tag{5}$$

where $s$ is a scaling factor that determines the sensitivity of the probability to the rating difference. Then, we can calculate any component $\Pr_B(i)$ (or $\Pr_{\neg B}(i)$ similarly) as the event that this model is chosen over each other model. This is calculated as:

$$\Pr_B(i) = \prod_j \Pr_B(i \text{ preferred over } j \mid \text{true Bradley-Terry coefficient ratings}) \tag{6}$$

For $\Pr_{\neg B}(i)$, the perturbed Bradley-Terry coefficient rankings are used instead.

# C. Experimental Details

## C.1. List of models

Table 6 lists the evaluated models and the methods used to query them. For all models, we rely on the default decoding hyperparameters (e.g., temperature) specified by the query method.

*Table 6.* Overview of evaluated models and the querying methods used in our experiments.

| Model | Company / Organization | Method of query in our experiments |
|---|---|---|
| claude-3-5-sonnet-20240620 | Anthropic | Anthropic API |
| claude-3-haiku-20240307 | Anthropic | Anthropic API |
| gemini-1.5-flash | Google | Google AI studio API |
| gemini-1.5-pro | Google | Google AI studio API |
| gemma-2-2b-it | Google | Together AI Inference API |
| gemma-2-9b-it | Google | Together AI Inference API |
| gemma-2-27b-it | Google | Together AI Inference API |
| gpt-3.5-turbo | OpenAI | OpenAI Text generation API |
| gpt-4-0125-preview | OpenAI | OpenAI Text generation API |
| gpt-4-1106-preview | OpenAI | OpenAI Text generation API |
| gpt-4-turbo-2024-04-09 | OpenAI | OpenAI Text generation API |
| gpt-4o-2024-05-13 | OpenAI | OpenAI Text generation API |
| gpt-4o-2024-08-06 | OpenAI | OpenAI Text generation API |
| gpt-4o-mini-2024-07-18 | OpenAI | OpenAI Text generation API |
| llama-3-8b-instruct | Meta | Together AI Inference API |
| llama-3-70b-instruct | Meta | Together AI Inference API |
| llama-3.1-8b-instruct | Meta | Together AI Inference API |
| llama-3.1-70b-instruct | Meta | Together AI Inference API |
| llama-3.1-405b-instruct | Meta | Together AI Inference API |
| mixtral-8x7b-instruct-v0.1 | Mistral AI | Together AI Inference API |
| mixtral-8x22b-instruct-v0.1 | Mistral AI | Together AI Inference API |
| qwen2-72b-instruct | Alibaba | Together AI Inference API |

## C.2. Prompts for embedding visualization

The three prompts we used for embedding visualization in Figure 2 are:

- Prompt #1: "Beside OFAC's selective sanction that target the listed individiuals and entities, please elaborate on the other types of US's sanctions, for example, comprehensive and sectoral sanctions. Please be detailed as much as possible"
- Prompt #2: "You are the text completion model and you must complete the assistant answer below, only send the completion based on the system instructions.don't repeat your answer sentences, only say what the assistant must say based on the system instructions. repeating same thing in same answer not allowed. user: descriptive answer for append many items to list python in python with proper code examples and outputs. assistant: "
- Prompt #3: "The sum of the perimeters of three equal squares is 36 cm. Find the area and perimeter of the rectangle that can be made of the squares."

## C.3. Details for the training-based detector

**Data collection and its cost.** The main cost of building the training-based detector comes from the data collection process, where the attacker gathers responses from various models for the same prompt and train classifier to distinguish among them (§ 2). In our experiments, we collect responses depending on the model type: For proprietary models, we directly used the model providers' APIs to obtain the responses. For open-source models, we relied on Together's API[7] to make the queries. We set the output length to 512 tokens and found that collecting 50 responses per model was sufficient to train an effective target model detector.

---

[7]https://www.together.ai/

To estimate the upper bound on the data collection cost, we based our calculations on the pricing of the most expensive model we tested. Proprietary models cost \$5.00 per 1 million output tokens, while open-source models cost \$1.80 per 1 million output tokens.

Using these rates, the upper bound cost of querying a single model is:

$$\text{Proprietary model: } 5.00 \times \frac{512 \times 50}{10^6} = 0.128 \quad \text{Open-source model: } \quad 1.80 \times \frac{512 \times 50}{10^6} = 0.046$$

Assuming the training process requires 10 proprietary models and 20 open-source models, the overall data collection cost would be approximately \$2.2 per prompt.

We collected data for 200 prompts in § 2, so the cost is at most \$440.

We use the logistic regression model from the scikit-learn library[8] with its default hyperparameters and a random state set to 42.

### C.4. Simulation Testbed

Our simulation in § 3 is based on an anonymized and deduplicated dataset of voting records from Chatbot Arena. The dataset includes 1,670,250 votes from 477,322 unique users, with 1,093,875 votes resulting in wins and 576,375 in ties. These votes cover 6,895 unique combinations of side-by-side model comparisons.

---

[8]`scikit-learn.org/1.5/modules/generated/sklearn.linear_model.LogisticRegression.html`

# D. More Experimental Results

## D.1. Target Model Detection

Table 7 presents the performance of identity-probing detector for all evaluated 22 models.[9]

*Table 7.* Averaged detection accuracy (%) with across 1,000 queries per prompt for different identity-probing prompts across various models.

| Model | Prompt | | | | |
|---|---|---|---|---|---|
| | Who are you? | Which model are you? | What is your model name? | How should I refer to you as an AI? | How would you define your role or identity as an AI? |
| claude-3-5-sonnet-20240620 | 99.3 | 100.0 | 98.5 | 100.0 | 100.0 |
| claude-3-haiku-20240307 | 100.0 | 96.3 | 100.0 | 42.9 | 14.3 |
| gemini-1.5-flash | 0.0 | 0.0 | 0.0 | 0.0 | 0.0 |
| gemini-1.5-pro | 97.2 | 96.5 | 100.0 | 0.0 | 99.1 |
| gemma-2-27b-it | 100.0 | 98.4 | 98.2 | 97.9 | 95.5 |
| gemma-2-2b-it | 81.8 | 91.8 | 58.2 | 12.7 | 4.5 |
| gemma-2-9b-it | 98.5 | 99.4 | 98.3 | 98.1 | 97.3 |
| gpt-3.5-turbo | 0.0 | 54.5 | 67.3 | 0.0 | 0.0 |
| gpt-4-0125-preview | 70.9 | 100.0 | 94.6 | 1.8 | 1.8 |
| gpt-4-1106-preview | 7.3 | 90.9 | 99.1 | 6.4 | 1.8 |
| gpt-4o-2024-05-13 | 16.4 | 93.3 | 99.9 | 0.0 | 6.4 |
| gpt-4o-2024-08-06 | 51.8 | 97.7 | 98.5 | 0.0 | 5.5 |
| gpt-4o-mini-2024-07-18 | 92.7 | 92.9 | 100.0 | 12.7 | 0.0 |
| llama-3-70b-instruct | 98.2 | 98.2 | 54.5 | 46.4 | 2.7 |
| llama-3-8b-instruct | 99.9 | 99.1 | 74.5 | 20.0 | 1.8 |
| llama-3.1-405b-instruct | 99.1 | 90.9 | 89.1 | 75.5 | 0.0 |
| llama-3.1-70b-instruct | 98.8 | 66.4 | 92.7 | 5.5 | 0.0 |
| llama-3.1-8b-instruct | 17.3 | 40.0 | 99.1 | 6.4 | 0.0 |
| mixtral-8x7b-instruct-v0.1 | 97.3 | 31.8 | 45.5 | 1.8 | 0.9 |
| mixtral-8x22b-instruct-v0.1 | 97.3 | 31.8 | 45.5 | 0.9 | 1.8 |
| qwen2-72b-instruct | 91.8 | 98.2 | 97.6 | 24.5 | 7.3 |

## D.2. Adversarial Vote

**Ablation for detector accuracy.** Table 8 shows the number of votes and interactions needed to shift a model's position by 1 to 50 places on the simulated leaderboard under different detector accuracies. As shown, the number of votes required to move a model up by 50 places increases by only about 150 when the detector accuracy drops from 1.0 to 0.9. This suggests that a detector, while not perfect, can still be sufficiently accurate to achieve the attack's objective.

**Ablation for non-detected actions.** When the attacker does not detect the target model, they can choose from four actions: randomly upvote one model, vote for a tie, vote both models as bad, or do nothing. The main results in § 3 assume the attacker does nothing. We also explore the other options in Table 9. As shown, there are no clear patterns indicating that any one option is significantly better than the others.

---

[9]We note that operation of these models was by University authors.

*Table 8.* The number of votes (a) and interactions (b) required to change the ranking of a low-ranked model on the simulated leaderboard, under varying detector accuracy.

| Target model=llama-13b (current rank: #129, #votes: 2443) | Target rank: 79 (↑ 50) | Target rank: 109 (↑ 20) | Target rank: 119 (↑ 10) | Target rank: 124 (↑ 5) | Target rank: 127 (↑ 2) | Target rank: 128 (↑ 1) |
|---|---|---|---|---|---|---|
| detector acc=1.0 | 1246 | 861 | 645 | 415 | 208 | 126 |
| detector acc=0.95 | 1304 | 918 | 682 | 522 | 255 | 126 |
| detector acc=0.9 | 1383 | 1012 | 732 | 525 | 271 | 136 |

(a) # Votes

| Target model=llama-13b (current rank: #129, #votes: 2443) | Target rank: 79 (↑ 50) | Target rank: 109 (↑ 20) | Target rank: 119 (↑ 10) | Target rank: 124 (↑ 5) | Target rank: 127 (↑ 2) | Target rank: 128 (↑ 1) |
|---|---|---|---|---|---|---|
| detector acc=1.0 | 80000 | 55000 | 40000 | 30000 | 15000 | 10000 |
| detector acc=0.95 | 85000 | 65000 | 45000 | 30000 | 15000 | 10000 |
| detector acc=0.9 | 100000 | 75000 | 55000 | 40000 | 20000 | 10000 |

(b) # Interactions

*Table 9.* The number of interactions required to change the ranking of a high-ranked model (a) and a low-ranked model (b) on the simulated leaderboard, under varying non-target strategies.

| Non-target strategy | Target rank: 1(↑ 4) | Target rank: 2(↑ 3) | Target rank: 3(↑ 2) | Target rank: 4(↑ 1) |
|---|---|---|---|---|
| Do nothing | 206000 | 184000 | 144000 | 18000 |
| Randomly upvote | 192000 | 182000 | 142000 | 16000 |
| Vote tie | 194000 | 182000 | 148000 | 20000 |
| Vote tie (both bad) | 196000 | 172000 | 152000 | 16000 |

(a) High-ranked model, claude-3-5-sonnet-20240620 (rank: #5)

| Non-target strategy | Target rank: 79 (↑ 50) | Target rank: 109 (↑ 20) | Target rank: 119 (↑ 10) | Target rank: 124 (↑ 5) | Target rank: 127 (↑ 2) | Target rank: 128 (↑ 1) |
|---|---|---|---|---|---|---|
| Do nothing | 80000 | 55000 | 40000 | 30000 | 15000 | 10000 |
| Randomly upvote | 75000 | 60000 | 40000 | 30000 | 15000 | 10000 |
| Vote tie | 80000 | 60000 | 40000 | 30000 | 15000 | 10000 |
| Vote tie (both bad) | 80000 | 60000 | 40000 | 30000 | 15000 | 10000 |

(b) Low-ranked model, llama-13b (rank: #129)

