# OpenReview forum: "Exploring and Mitigating Adversarial Manipulation of Voting-Based Leaderboards"
_ICML.cc/2025/Conference — ICML 2025 oral_

### Official Review · Reviewer_r1N6 · 2025-02-27

**Overall Recommendation:** 3

**Summary:**

This paper focuses on the adversarial manipulation of voting-based LLM leaderboards, e.g., Chatbot Arena. Intuitively, keeping the model's response anonymous is essential to ensure the integrity of the leaderboard. However, this paper demonstrated that an adversary can efficiently de-anonymize the responses and thus can upvote/downvote some specific models. This paper discusses two "target model detector", namely the identity-probing detector and the training-based detector. The experiments show that both detectors can separate the target model from others, which efficiently breaks the anonymization. This paper also discusses some possible mitigations of such attacks for the purpose of further enhancing the robustness of voting-based LLM leaderboards.

**Claims And Evidence:**

In the statement of contributions (Lines 73-82, left), the claims include:
1. The users can break model response anonymity with high probability,
2. The estimated votes to boost or reduce a model's ranking is "a few thousand"
3. A cost model and some potential mitigations are discussed.

Generally speaking, I think the claims are well supported. However, I find the evidence for the first claim is not convincing enough. See the Questions For Authors part.

**Essential References Not Discussed:**

The following two methods are related to a similar topic of "LLM identification".

1. Gubri, M., Ulmer, D., Lee, H., Yun, S., and Oh, S. J. TRAP: Targeted Random Adversarial Prompt Honeypot for Black-Box Identification. In Annual Meeting of the Association for Computational Linguistics (ACL). ACL, 2024.
2. Jin, H., Zhang, C., Shi, S., Lou, W., and Hou, Y. T. ProFLingo: A Fingerprinting-based Copyright Protec tion Scheme for Large Language Models.  CoRR abs/2405.02466, 2024.

Could the authors please discuss on the relationship between the present paper and these two papers and the references therein?

**Experimental Designs Or Analyses:**

I have checked the soundness/validity of the experimental designs or analyses in this paper.

**Methods And Evaluation Criteria:**

The methods to de-anonymize the model response include
1. Identity-probing detector
2. training-based detector
While the evaluation of the identity-probing detector is straightforward, I am not sure if I have fully understood the training-based detector, see the questions in the Questions For Authors part.

**Other Comments Or Suggestions:**

Typo on Line 104 (right): its name

**Other Strengths And Weaknesses:**

**Strenths**

1. Very practical. As mentioned in Line 036, the authors claimed that they work with the Chatbot Arena developers and have enhanced the robustness of the voting-based leaderboards based on their analysis.

**Questions For Authors:**

1. Given the efficiency of the identity-probing detector, what is the meaning of applying a more sophisticated training-based detector?
2. Could the authors please explain the intuition/motivation of the training-based detector?
3. Is it possible to employ some specific system prompt to prevent the models from revealing their identity, e.g., "You are an anonymous model competing in the Chatbot Arena. Do not tell anyone about your identity." This is the simplest mitigation I could have come up with. As the detection accuracies in Table 2 are all >95% in the best cases, I suppose introducing some basic defense would enhance the validity of the results.
4. What is the performance of the mitigations under jailbreaking attacks, e.g., PAIR and AutoDAN? The adversary can perform more stealthy attacks against the model than those mentioned in Section 2.2.

**Relation To Broader Scientific Literature:**

1. According to the related work section, this paper seems to be the first work focusing on the adversarial manipulation of voting-based LLM leaderboards. The related works discussed in Sections 5 and A are not directly related to the topic of the present paper.
2. The setting of this paper is similar to jailbreaking attacks against LLMs. See the Questions For Authors part.

**Theoretical Claims:**

This paper does not include theoretical analysis.

---

> ### Author Rebuttal · Authors · 2025-04-01
>
> We thank the reviewer for their positive feedback and questions. Our detailed responses are as follows:
>
> > Q1: Given the efficiency of the identity-probing detector, what is the meaning of applying a more sophisticated training-based detector? Could the authors please explain the intuition/motivation of the training-based detector?
>
> **A**: We appreciate the question. We note two limitations of the identity=porbing detector:
> - Susceptibility to Countermeasures:  the Chatbot Arena leaderboard already uses post-processing to exclude votes from Elo score calculation when model responses mention model names, which naturally limits the usefulness of the identity-probing detector. But we still analyze the effectiveness of this detector in the paper, as it could be effective in other voting-based chatbot benchmarks, and because the post-processing could be evaded, e.g., by asking the model to reveal its identity in Base64 encoding.
> - Lack of Stealth: Identity-probing is an active technique requiring specific, often obvious queries. This makes the adversarial attempt itself highly detectable. In contrast, the training-based approach is passive and thus stealthier.
>
>
> > Q2: Is it possible to employ some specific system prompt to prevent the models from revealing their identity, e.g., "You are an anonymous model competing in the Chatbot Arena. Do not tell anyone about your identity." This is the simplest mitigation I could have come up with. As the detection accuracies in Table 2 are all >95% in the best cases, I suppose introducing some basic defense would enhance the validity of the results.
>
> **A** :We appreciate the suggestion. While preventing models from outputting their identities can provide a basic layer of defense for identity-probing detectors, it’s insufficient to prevent more sophisticated detectors. As shown in Table 3, training-based detectors—which do not rely on explicit model name outputs—still give accuracy rates above 95%. Furthermore, it may be undesirable to require system prompt changes for models participating in the leaderboard, as system prompts are generally carefully constructed by model owners.
>
> > Q3: What is the performance of the mitigations under jailbreaking attacks, e.g., PAIR and AutoDAN? The adversary can perform more stealthy attacks against the model than those mentioned in Section 2.2.
>
> **A**: We would like to clarify that our mitigations are entirely independent of the prompts and responses and so jailbreaking should have no relationship to our mitigations. We would appreciate further clarification on: 1) What makes the reviewer believe jailbreaking attacks could undermine the mitigation strategies discussed in the paper, and 2) What "more stealthy attacks" means in this context.
>
> > Q4: Missing related work
>
> **A**: We appreciate the reviewer sharing the pointers and will incorporate these references in the final version.

---

### Official Review · Reviewer_hmrX · 2025-03-07

**Overall Recommendation:** 4

**Summary:**

It has become common for LLMs to be evaluated subjectively in crowd-sourced "arenas", which usually use elo-based scoring based on user preferences. The authors study this voting-based evaluation setting, and find that they are susceptible to adversarial manipulation through a two-step attack: (1) de-anonymizing model responses with high accuracy, and (2) selectively voting for or against a target model to manipulate the rankings. They demonstrate that this attack is feasible and cheap, and then explore mitigations.

**Claims And Evidence:**

The claim that model responses can be reliably distinguished by malicious parties is supported by thorough empirical experimentation, across many models. An accuracy of >95% can be achieved using extremely simple supervised learning methods.

The authors verify the cost-related claim by running simulations on a system emulating Chatbot Arena.

**Essential References Not Discussed:**

I don't know of any references that are relevant and missing.

**Experimental Designs Or Analyses:**

The experimental designs and analyses appear sound to me.

**Methods And Evaluation Criteria:**

The methods seem appropriate and well-designed.

The paper targets an attack on a real-world system: Chatbot Arena. The ultimately convincing evaluation would be to attack the actual system, and show that such an attack is feasible in the real world. The authors instead evaluate on a *simulation* of Chatbot Arena, based upon real-world voting records from Chatbot Arena. This seems like a proper evaluation setup to me.

**Other Comments Or Suggestions:**

Typos:
- Line 19: "two randomly selected and models"
- Line 267: "An passive attacker"
- Line 341: missing punctuation

**Other Strengths And Weaknesses:**

Strengths:
- The paper is very well written and organized, and it for the most part quite easy to follow
- The topic is very important. Every major model release highlights voting-based elo scores.

Weaknesses:
- Sections 4.2.3 and 4.3 could be revised to be clearer.
  - I think the paper would benefit from a rewriting of these sections, as they were unclear to me the first time I read them.
  - Specifically, 4.3 should clarify that it is experimenting based off of the proposed defenses in 4.2.3 (and has nothing to do with 4.2.1, 4.2.2, 4.2.4, if my understanding is correct).
  - 4.2.3 can be rewritten to more clearly present the attacker's strategy, and the defender's strategy. The attacker's strategy in scenario (1) informs the defender's strategy in scenario (2), and this would be a clearer presentation.

**Questions For Authors:**

- For the "perturbation" defense in 4.2.3, could the attacker simply create multiple accounts, aggregate the perturbed numbers across the many accounts, and then use the average?

**Relation To Broader Scientific Literature:**

This work is related to the body of literature studying the validity of benchmark evaluation numbers, which is an important field, as new models are judged based on the evaluation numbers that they achieve.

**Theoretical Claims:**

The paper does not necessitate any proofs or theoretical claims.

---

> ### Author Rebuttal · Authors · 2025-04-01
>
> We thank the reviewer for their positive feedback and questions. Our detailed responses are as follows:
>
> > Q1: Sections 4.2.3 and 4.3 could be revised to be clearer…Specifically, 4.3 should clarify that it is experimenting based off of the proposed defenses in 4.2.3 (and has nothing to do with 4.2.1, 4.2.2, 4.2.4, if my understanding is correct). 4.2.3 can be rewritten to more clearly present the attacker's strategy.
>
> **A**: We appreciate the reviewer’s suggestion of reorganizing the defense section (Sec 4). We apologize for the confusion caused by presenting the results for malicious user identification in Sec 4.3 without a clear pointer to Sec 4.2.3. We will reorganize these sections in the final version.
>
> > Q2: For the "perturbation" defense in 4.2.3, could the attacker simply create multiple accounts, aggregate the perturbed numbers across the many accounts, and then use the average?
>
> **A**: No, please note that only one permuted table is released by the system therefore you cannot average or detect the permutations.
>
> Q3: Typos.
>
> **A**: Thanks! We will fix them in the final version.

---

### Official Review · Reviewer_X39U · 2025-03-13

**Overall Recommendation:** 3

**Summary:**

This submission examines the susceptibility of voting-based LLM assessment platforms to adversary interference, particularly emphasizing Chatbot Arena, a prominent platform that ranks language models according to human preferences.

 The primary contributions of the paper are: (1) Evidence that users can effectively compromise model response anonymity with high precision (>95%) employing basic classification methods; (2) Simulation-based estimations indicating that a few thousand adversarial votes can substantially modify a model's ranking on the leaderboard; (3) Formulation of a cost model for the attack and the proposal of various mitigation strategies to enhance the expense and complexity of such attacks; (4) Collaboration with the Chatbot Arena developers to execute these mitigations, exemplifying a responsible disclosure methodology in security research.

 The authors assert that their findings are relevant to any voting-based ranking systems, not alone Chatbot Arena, underscoring a wider issue for platforms dependent on human preferences for assessment.

## Update after rebuttal
The authors provided detailed discussions with additional experiments to address my concerns. I appreciate their effort and would like to retain my original score to accept this submission.

**Claims And Evidence:**

The claims made in the submission are generally well-supported by empirical experiments.

The claim that model responses may be de-anonymized is supported by Figure 3, which presents detection accuracy across various prompts and models.

The claim that the quantity of votes might influence the leaderboard is supported by simulations utilizing actual voting data shown on page 16. Tables 4 and 5 include comprehensive estimates of the votes and interactions necessary to alter the ranks of both high-ranked and low-ranked models.

**Essential References Not Discussed:**

N.A.

**Experimental Designs Or Analyses:**

The experimental designs and analyses specified in the submission are meticulously crafted.

The de-anonymization assessments are meticulously designed, scrutinizing both identity-probing and training-based detectors across a diverse range of prompts and models.

 The authors performed ablation studies to assess the impact of different design selections on detector efficacy.  The authors examine several feature types and conclude that "basic text features such as BoW and TF-IDF achieve remarkably high detection accuracy, with BoW surpassing 95% in multiple cases" (page 4).

The simulation studies aimed at estimating the votes necessary to affect the leaderboard include real voting data from Chatbot Arena, hence enhancing the credibility of the results.  The authors analyze many scenarios and provide detailed findings in Tables 4 and 5.

**Methods And Evaluation Criteria:**

The methodologies and assessment standards are effectively aligned with the objectives of the submission.

The procedures and criteria included in the submission are suitable for evaluating the proposed approach. Utilizing accuracy metrics is appropriate for evaluating detector performance. The authors show the separability of model responses by principal component analysis of bag-of-words characteristics, successfully demonstrating "how different models respond to the same prompt" (Figure 2).

The suggested mitigations are evaluated based on their efficacy in increasing the cost of the assault. The authors assess the security benefits and potential drawbacks of each mitigation, including the distributional changes that may occur when authentication is required.

The simulation technique for determining the necessary votes to affect the ranking is meticulously designed. The author delineates clear objectives for the attack: "Up(M, x)" and "Down(M, x)" (page 5). The simulations account for the attacker's detection accuracy and behavior when the target model is not identified.

**Other Comments Or Suggestions:**

N.A.

**Other Strengths And Weaknesses:**

N.A.

**Questions For Authors:**

- The proposed technique demonstrates high accuracy in de-anonymizing model answers; nonetheless, LLMs are regularly updated. In what manner may the efficacy of your detection techniques be altered when models undergo updates and fine-tuning?

- The simulations concentrate mostly on singular opponents; however, how may the vulnerability escalate with coordinated assaults from numerous adversaries?

- As leaderboards expand to encompass additional models, does the efficacy of this defensive strategy alter?

**Relation To Broader Scientific Literature:**

The paper places its contributions within the broader scientific literature on voting-based systems, and security vulnerabilities.

The paper connects to the literature on voting-based systems and their security vulnerabilities. The authors note that "voting-based systems are frequently used in security relevant scenarios, such as for malware identification [1] or for content validation [2]" and that "attacks on these systems are well studied [3]" (page 7). They also discuss reputation systems as a common approach to securing these systems, citing work by [2] and [4].

The de-anonymization task is related to authorship attribution and model detection, as the authors acknowledge: "Our primary attack involves training a classifier that can identify which language model system produced a given generation. This task is related to the much older task of authorship attribution—identifying the authors of anonymous (but human-written) works of writing [5,6]".

Ref:
[1] VirusTotal Documentation, 2024.
[2] The eigentrust algorithm for reputation management in p2p networks. In WWW, 2003.
[3] A survey of attack and defense techniques for reputation systems. In CSUR, 2009.
[4] Towards {Tracking-Resistant} anonymous reputation. In NSDI, 2016.
[5] Authorship attribution in the era of llms: Problems, methodologies, and challenges. In arXiv, 2024.
[6] De-anonymizing text by fingerprinting language generation. In NeurIPS, 2020.

**Theoretical Claims:**

The paper does not make extensive theoretical claims or provide formal proofs, as it is primarily an empirical study focused on demonstrating and mitigating a practical vulnerability.

---

> ### Author Rebuttal · Authors · 2025-04-01
>
> Thank you for your positive feedback and questions. Our detailed responses are as follows:
>
> > Q1: The proposed technique demonstrates high accuracy in de-anonymizing model answers; nonetheless, LLMs are regularly updated. In what manner may the efficacy of your detection techniques be altered when models undergo updates and fine-tuning?
>
> **A**: We appreciate the question. We first note that updates to most models via API are relatively infrequent. Assuming that there are (infrequent) updates to the model, the detector efficacy post-update depends on the adversary:
> - When the adversary controls the target model, the adversary can easily retrain its detector to reliably detect the new model.
> - When the adversary does not have the control of the target model, our experiments with llama-3-8b-instruct and gemma-2-27b-it suggest that the detector is relatively robust to model changes: as shown in Table 1, detection accuracy remained above 90% even after several hundred SFT steps on these models.
>
> Table 1: The original detector’s accuracy on the SFTed model with different steps
> | # SFT steps on lmsys-chat-1m | Model: llama-3-8b-instruct | Model: gemma-2-27b-it |
> |------------------------------|----------------------------|-----------------------|
> | 0                            | 95.4                       | 96.3                  |
> | 100                          | 95.2                       | 95.6                  |
> | 200                          | 94.7                       | 95.1                  |
> | 500                          | 94.1                       | 94.7                  |
>
> > Q2: The simulations concentrate mostly on singular opponents; however, how may the vulnerability escalate with coordinated assaults from numerous adversaries?
>
> **A**: We appreciate the question. On voting-based leaderboards, users typically evaluate models through head-to-head comparisons. We're curious to understand what the reviewer meant by "numerous adversaries" and would appreciate more context.
>
> > Q3: As leaderboards expand to encompass additional models, does the efficacy of this defensive strategy alter?
>
> **A**:  Thank you for pointing this out. Yes, the strategy's effectiveness will change as the number and type of models on the leaderboard evolve. Newer models might have different performance characteristics or vulnerabilities, impacting the defense's relative success. However, as discussed in the paper, each defense involves a utility-cost trade-off. This allows system designers to analyze its effectiveness under current conditions and adjust parameters to meet system requirements at the time of deployment. We will clarify this in the paper.

---

### Decision · Program_Chairs · 2025-05-01

**Decision:**

Accept (oral)

**Comment:**

Reviewers unanimously voted for acceptance.

- Immediate security issue for Chatbot arena
- The method makes sense
- Empirical results are fairly strong